# Non-Coding RNAs Are Implicit in Chronic Myeloid Leukemia Therapy Resistance

**DOI:** 10.3390/ijms232012271

**Published:** 2022-10-14

**Authors:** Alexander Rudich, Ramiro Garzon, Adrienne Dorrance

**Affiliations:** Comprehensive Cancer Center, The Ohio State University, Columbus, OH 43210, USA

**Keywords:** chronic myeloid leukemia, non-coding RNA, lncrna, circRNA, miRNA, therapy resistance, imatinib, tyrosine kinase inhibitor, leukemia stem cell, dysregulation

## Abstract

Chronic myeloid leukemia (CML) is a myeloproliferative neoplasm initiated by the presence of the fusion gene *BCR::ABL1*. The development of tyrosine kinase inhibitors (TKIs) highly specific to p210*^BCR^*^-*ABL1*^, the constitutively active tyrosine kinase encoded by *BCR::ABL1*, has greatly improved the prognosis for CML patients. Now, the survival rate of CML nearly parallels that of age matched controls. However, therapy resistance remains a persistent problem in the pursuit of a cure. TKI resistance can be attributed to both *BCR::ABL1* dependent and independent mechanisms. Recently, the role of non-coding RNAs (ncRNAs) has been increasingly explored due to their frequent dysregulation in a variety of malignancies. Specifically, microRNAs (miRNAs), circular RNAs (circRNAs), and long non-coding RNAs (lncRNAs) have been shown to contribute to the development and progression of therapy resistance in CML. Since each ncRNA exhibits multiple functions and is capable of controlling gene expression, they exert their effect on CML resistance through a diverse set of mechanisms and pathways. In most cases ncRNAs with tumor suppressing functions are silenced in CML, while those with oncogenic properties are overexpressed. Here, we discuss the relevance of many aberrantly expressed ncRNAs and their effect on therapy resistance in CML.

## 1. Introduction

Chronic myeloid leukemia (CML) is a myeloproliferative disorder cytogenetically recognizable by the presence of the Philadelphia (Ph) chromosome. The formation of the Ph chromosome is a result of a reciprocal translocation between the long arms of chromosomes 9 and 22 t (9;22) (q34;q11) [1]. The fusion of the *Breakpoint Cluster Region* (*BCR*) and *Abelson Tyrosine-Protein Kinase 1* (*ABL1*) genes resulting from the translocation generates the chimeric oncogene *BCR::ABL1* that is both necessary and sufficient for the development of CML [2,3,4,5,6,7,8,9]. In the absence of this fusion, *ABL1* encodes a ubiquitously expressed non-receptor tyrosine kinase which regulates cell cycle progression, proliferation, DNA-repair, and differentiation in response to a variety of intracellular and extracellular signals including integrin signaling, cytokines, and genotoxic stress [2,10,11,12]. When fused with *BCR*, the activity of *ABL1* is no longer dependent on these signals and becomes the constitutively active tyrosine kinase p210*^BCR^*^-*ABL1*^ [9,13,14]. In addition to CML, p210*^BCR^*^-*ABL1*^ transcripts have also been rarely detected in Acute Lymphocytic Leukemia (ALL) and Acute Myeloid Leukemia (AML) [15,16]. p210*^BCR::ABL1^* is the most common product of the fusion in CML but a shorter transcript p190*^BCR::ABL^* is also detected in CML, although more commonly in AML and B-cell ALL (B-ALL) [15,16]. A third variant p230*^BCR::ABL1^* is the major diagnostic marker for Neutrophilic-chronic myeloid leukemia (CML-N) [17].

The result of *BCR::ABL1* expression is the transformation of the hematopoietic stem cell (HSC) into the leukemic stem cell (LSC) which initiates and propagates CML [18]. Although the presence of *BCR::ABL1* is sufficient to cause the transformation of the HSC to the LSC and initiate CML, many other events contribute to the development and persistence of this disease.

The generation of CML is not a one-step process. The progression of the disease occurs in three phases: the chronic phase (CP), the accelerated phase (AP), and the blast phase (BP). Each phase is classified by the percentage of blasts in the bone marrow or blood. As the disease progresses it becomes more resistant to therapy and associates with poor survival. Patients who are diagnosed with CP-CML are more manageable and are very responsive to current targeted therapies, while patients diagnosed with BP-CML have most aggressive disease, equivalent to acute leukemia. If left untreated, CML will progress to the BP in just a period of three to eight years [19].The presence of the Ph chromosome is sufficient to initiate the CP-CML but not the BP-CML. Progression to BP-CML requires acquisition of a variety of additional mutations. In addition to protein coding gene mutations, aberrant ncRNAs expression also contributes to progression of the disease.

The current first-line standard of care for CML patients is treatment with tyrosine kinase inhibitors (TKIs) that competitively or allosterically inhibit p210*^BCR^*^-*ABL1*^, as well as other *BCR::ABL1* variants, although less effectively [2,20,21,22]. Currently, there are three generations of TKIs available, but the first generation TKI Imatinib (IM) is typically preferred due to the availability of more affordable generics. Additionally, because second generation TKIs have higher affinity compared to IM, they usually associate with long-term toxicities, making them less desirable as a first line treatment [23].

Patients treated in the chronic phase of CML have a life expectancy approaching those of age matched controls [24]. Because of the success of TKI treatment, a major goal for CML patients is to achieve treatment free remission (TFR). Recent studies demonstrate that patients with undetectable residual disease for a period of two years can safely attempt to discontinue treatment with close molecular monitoring for disease [25,26,27]. Of patients who maintained undetectable residual disease for at least one year and chose to discontinue treatment, 51.1% experienced molecular relapse-free remission after five years, with length of TKI administration being a major predictor of outcome [26]. The efficacy of treating CP-CML with TKIs is clear and the potential for TFR is promising for many patients and should continue to be the focus of future research.

However, despite the undisputed success of TKIs in treating many CML patients, there are a few downsides to this therapy. Some patients are inherently unresponsive to first line TKI therapy or develop resistance during treatment. Of those patients, 40–60% possessed at least one mutation in the kinase domain which is responsible for the drug resistance [28,29,30]. While kinase domain mutations are the most common mechanism of therapy resistance in CML, *BCR::ABL1* amplification is also common and can overtake the inhibitory capacity of TKIs by simply out numbering the drug [31]. Additionally, *BCR::ABL1* independent mechanisms contribute to therapy resistance by increasing genomic instability, modifying drug transporters, changing the bone marrow microenvironment (BMM), supporting LSCs, and activating signaling pathways which promote survival [32,33,34]. In these cases, the implementation of second generation TKIs greatly reduces the risk of the disease progressing to the AP-CML or BP-CML [2,35,36]. Unfortunately, because of their increased potency, the use of second generation TKIs often carries severe negative side effects such as increased incidence of vascular defects, pleural effusions, and cardiovascular disease, preventing their use in many patients [37,38,39,40,41,42,43,44].

CML LSCs also provide a therapeutic challenge as they are not dependent on *BCR::ABL1* for survival and therefore cause relapse after treatment discontinuation [45]. Furthermore, LSCs are heterogeneous, and mutations acquired in the LSC can give them a survival advantage, leading to the evolution of TKI resistant clones and progression to BP-CML [46,47]. *BCR::ABL1*, therapy resistance, and CML LSCs must be better understood and targeted to achieve a cure for CML.

Recently, many studies have focused on the role of ncRNAs in CML. ncRNAs contribute to therapy resistance and disease progression through a variety of mechanisms because they can exhibit multiple different functions and regulate gene expression responsible for many biological processes. In this review we explore the most studied ncRNAs in CML and their role in contributing to therapy resistance: miRNAs, circRNAs, and lncRNAs.

## 2. The Role of miRNAs in CML

miRNAs are small RNA molecules with a length between 18–22 nucleotides that regulate gene expression by mainly preventing translation of transcribed messenger RNA (mRNA). miRNAs accomplish this generally by binding to the 3′ untranslated region (UTR) of mRNA and associating with RNA-induced silencing complex (RISC) to degrade its complementary target mRNA. Their complementary pairing with mRNAs can also prevent their localization and association with ribosomes, even in the absence of the RISC complex. Because they can regulate gene expression, miRNAs control a vast array of biological processes including differentiation, proliferation, and apoptosis which explains why a variety of miRNAs have been implicated in many different malignancies. CML is no exception, as a variety of miRNAs are shown to be dysregulated, contributing to the disease. The development of miRNA mimics, which can be used to restore miRNA expression in cases in which it was downregulated, and antagomirs, which may be used to mitigate overexpression of mRNAs, demonstrate that miRNA dysregulation is a targetable aberration. Currently, the primary methods of delivery of these treatments are viral or lipid nanoparticles but each method has limitations regarding its translation into the clinic. Despite this, further characterization of the functions of miRNAs in CML is crucial, so that once delivery methods are improved, miRNA therapies can be readily deployed. Furthermore, although miRNAs are small, single-stranded RNAs, it is reported that they are relatively stable and may be useful as potential biomarkers for a variety of diseases [48]. Here, we demonstrate that a variety of mechanisms are responsible for the dysregulation of miRNAs, including BMM remodeling, epigenetic modifications, and activation or suppression of pathways in response to increased *BCR::ABL1* activity. The aberrantly expressed miRNAs then exert their effect by targeting *BCR::ABL1*, CML LSCs, pathways implicit in therapy resistance, and drug transporters.

### 2.1. BCR::ABL1 Is a Target of miRNAs

Because the development of CML is a direct result of the fusion protein transcribed from *BCR::ABL1*, miRNAs which directly target these transcripts before they can be translated into the oncoprotein are obvious therapeutic candidates and were the focus of early studies of non-coding RNAs in CML. Several miRNAs targeting *BCR::ABL1* transcripts have been identified and characterized. Since miR-29b is downregulated in CML patients, it was selected for further investigation showing that overexpression of miR-29b in the K562 CML cell line leads to a decrease of *BCR::ABL1* expression. The 3′ UTR of *BCR::ABL1* was confirmed as the target site for miR-29b using a luciferase assay and overexpression of miR-29b induced apoptosis and slowed proliferation in vitro [49]. Similar findings occurred when researchers investigated miR-30a and miR-424, two other lowly expressed miRNAs in the bone marrow of CML patients [50,51]. However, they were also shown to play a role in IM resistance. Overexpression of miR-424 was shown to sensitize K562 cells to IM treatment [51]. IM treatment of CML decreases miR-30a expression. Since miR-30a represses its targets Beclin 1 and ATG5, which promote intrinsic apoptosis, the decrease of miR-30a following treatment with IM is likely a contributor to resistance [52]. These miRNAs both directly reduce *BCR::ABL1* expression, and their down-regulation contributes to IM resistance. Therefore, reestablishing homeostatic levels of miR-424 and miR-30a via introduction of miRNA mimics may be an effective strategy to mitigate *BCR::ABL1* overexpression that contributes to disease severity and therapy resistance [52].

Since we know that the BMM and niches play a significant role in contributing to therapy resistance, it is likely that miRNAs are dysregulated there as well. Indeed, like the previously discussed miRNAs, miR-320a regulates *BCR::ABL1* and was decreased in CML mesenchymal stromal cells (MSCs) [53]. Low miR-320a levels were correlated with decreased survival rates of CML patients [53]. Restoring baseline levels of miR-320a can slow disease progression by limiting phosphorylation of the PI3K/AKT pathway, downstream of *BCR::ABL1*, which reduces growth and survival of CML and has been shown to sensitize CML LSCs to IM [53,54]. Delivery of miR-320a mimics to the leukemic niche is a potential strategy to address its aberrant expression in MSCs.

Since current miRNA mimic delivery methods have limitations, other therapies to address aberrant expression of miRNAs should be explored. In fact, multiple studies have elucidated epigenetic modifications which contribute to the silencing of tumor suppressing miRNAs. Methylation of CpG islands, specifically, is a useful tool for cancer progression, as it allows tumor suppressors to be efficiently silenced. miR-196b was shown to be heavily methylated in CML patients, explaining its limited expression in CML patients when compared to healthy controls [55]. Luciferase assays designated both *BCR::ABL1* and HOXA9 oncogenes as targets of miR-196b and further in vitro experiments confirmed miR-196b could suppress them, diminishing proliferation and cell cycling [55]. Similar data indicates that miR-23a is hypermethylated in CML, which coordinates with its decreased expression in CML patient samples and cell lines [56]. Expression of miR-23a was inversely correlated with that of *BCR::ABL1* and overexpression of miR-23a decreased *BCR::ABL1* inducing apoptosis [56]. These results implicate miR-196b and miR-23a as tumor suppressors, which are epigenetically silenced to further the progression of CML by increasing the translation of *BCR::ABL1*. Introducing hypomethylating agents could potentially restore their functions, reducing disease progression and resistance. Indeed, the use of hypomethylating agents to treat CML has been documented, with recent data indicating a synergistic effect of an oral demethylating agent (OR-2100) with TKIs on CML growth in vitro [57]. Another study demonstrated some success in treating IM resistant patients with low doses of decitabine [58]. It was also demonstrated that the combination of administration of low doses of decitabine along with IM is effective in some patients with advanced stages of CML [59].

Like previously discussed miRNAs, miR-96 directly targets the 3′ UTR of *BCR::ABL1* and ABL1, however, the suppression of miR-96 is closely associated with the transition from CP to BP [60]. The downregulation of miR-96 leads to an increase in *BCR::ABL1* products, progressing the disease [60]. The utility of miR-96 may extend beyond therapeutic promise, as the close association of miR-96 expression levels to disease stage may make it a useful biomarker to track the molecular progression of CML.

### 2.2. miRNAs Regulating CML-Leukemia Stem Cells

LSCs not only initiate but also contribute to disease progression, persistence, and relapse in CML patients. However, LSC quiescence and independency of *BCR::ABL1* make them a particularly challenging target [45,46,47]. The dysregulation of several miRNAs has been shown to contribute to therapy resistance in CML by facilitating the persistence of the LSC.

Let-7, an miRNA precursor, is implicated as a tumor suppressor in a variety of solid tumors including lung, ovarian, breast, and colorectal cancer [61]. Recent data suggests let-7, is dysregulated in the blast crisis of CML. Due to the increase in JAK2, which occurs during blast crisis because of an increase of *BCR::ABL1*, adenosine deaminase acting on RNA1 (ADAR1) is stimulated. The newly activated ADAR1 modifies the early precursor pri-let-7 preventing the biosynthesis of let-7. This blocks its ability to give rise to mature miRNAs and promotes an increase in LIN28B, a transcription factor promoting pluripotency, known to contribute to cancer progression [62]. The decrease of let-7 and subsequent increase in LIN28B contributes to the self-renewal capacity and stem cell status of the LSC, ultimately contributing to relapse and therapy resistance [63]. Downregulation of miR-203 was also shown to contribute to the persistence of CML LSCs by controlling their growth and self-renewal. It acts by targeting the 3′ UTRs of survivin and Bmi-1 transcripts [64]. Survivin is known to play an important role in preventing apoptosis, while Bmi-1 is linked to stem cell self-renewal properties [65,66]. By inhibiting these molecules, miR-203 prevents the survival and renewal of LSCs, combatting disease persistence and relapse; however, its dysregulation in CML enables LSC persistence [64]. Furthermore, miR-494-3p is decreased in CML LSCs and increases competency of all three generations of TKIs to induce apoptosis. It binds to the 3′ UTR of *c-MYC,* limiting its translation and exerting a pro-apoptotic effect that is independent of *BCR::ABL1* [67]. Since these miRNAs help suppress stemness in CML or sensitize the stem cell to TKI treatment, restoring their expression in patients using miRNA mimics is a potential therapeutic strategy in combination with traditional therapy. These combination of treatments addresses not only the oncoprotein of *BCR::ABL1*, but also one of the most important *BCR::ABL1* independent mechanisms of therapy resistance and relapse: the survival and self-renewal of the LSC.

Conversely to Let-7, miR-203, and miR-494, many miRNAs exhibit oncogenic properties and so they are often upregulated in CML. Up-regulated in many human cancers, miR-21 has been shown to contribute to oncogenesis and targeting miR-21 has led to an increase in efficacy of a variety of therapies [68]. In CML, data show that miR-21 protects the CML LSC following treatment with IM. Concurrent treatment with antogmiR-21 and IM increased the efficacy of IM and apoptosis of CD34+ CML cells without affecting normal CD34+ cells [54].The authors concluded that antagonizing miR-21 increases efficacy of IM on LSCs by targeting the PI3K/AKT pathway. Because the PI3K/AKT pathway is necessary for essential function of many stem cells, it should not be directly targeted [69]. Thus, antagomir-21 in combination with IM treatment is a creative way to target increased PI3K/AKT signaling, without disrupting normal stem cell function [54].

miR-29a-3p, which is also upregulated in CML LSCs, suppresses *TET2* by binding to its 3′ UTR. *TET2* contributes to the apoptotic response in response to TKI treatment and has been shown to act as a tumor suppressor, particularly for malignancies of the myeloid descent [70]. Therefore, antogomiR-29a-3p may alleviate therapy resistance by silencing its target, restoring *TET2* transcript translation, and apoptosis in response to TKI administration [67]. Similarly, miR-660-5p desensitizes CML to IM and is up-regulated in CML LSCs. miR-660-5p protects CML LSCs by binding the 3′ UTR of endothelial PAS domain protein 1 (*EPAS1*), mitigating its function in regulating cellular responses to hypoxic environments [67]. miR-378, increased in bone marrow from CML patients, is shown to increase expression of multiple stem cell markers including Nanog, Oct4, and c-MYC [71]. Previous studies have linked these transcription factors to leukemias of the myeloid lineage [72,73]. Researchers further characterized the oncogenic functions of miR-378 by validating that FUS1, a tumor suppressor, is a target of miR-378 in K562 cell lines; thus, miR-378 induces proliferation of CML [71]. Taken together, we can conclude that miR-378 promotes self-renewal and the pluripotency of LSCs in CML, contributing to disease persistence and therapy resistance.

miR-126-3p (miR-126) contributes to disease persistence by regulating LSC dormancy and engraftment potential. Although miR-126 is downregulated in the LSC compared to healthy long-term HSCs, it is upregulated in the leukemic niche, particularly in endothelial cells which secrete miR-126 to the LSCs [74]. In addition, TKI treatment leads to an increase in miR-126 further contributing to LSC persistence and stemness. Data shows that inhibition or deletion of miR-126 in mice led to improved response to TKIs and reduced LSC derived relapse [74]. Returning these dysregulated miRNAs to their homeostatic levels using antagomiRs could greatly improve CML LSC response to TKIs, allowing patients to cease TKI treatment after shorter periods of time and see lower rates of LSC induced relapse.

Aside from the treatment challenges presented by LSCs, impaired differentiation is an established obstacle in both AML and BP CML [75,76]. While most of the miRNAs discussed thus far exert their effects by inhibiting the translation of mRNAs, miR-328 has been shown to operate outside of this classical description of miRNAs. In addition to binding mRNAs to prevent their translation, miR-328 is also capable of interacting with the heterogenous ribonucleoprotein (hnRNP) E2 [77]. In general, hnRNPs interact with DNA and RNA to yield a properly transcribed, modified, and translated mRNA product [78]. However, they may also bind to mRNA and prevent their translation [79]. In CML BP, where abundant *BCR::ABL1* transcripts are generated, an increase in MAPK activation yields more hnRNP E2, ultimately causing a block in differentiation by binding to *CEBPA* and preventing its translation [80]. Interestingly, miR-328 is also downregulated in BP via a *BCR::ABL1* dependent pathway [77]. It was shown that the absence of miR-328 caused a decrease in the generation of *CEBPA* protein, as under normal conditions, miR-328 competes with *CEBPA* mRNA for binding to hnRNP E2, releasing *CEBPA* transcripts for translation [77]. Expectedly, ectopic expression of miR-328 restored *CEBPA* translation [77]. Although this study of miR-328 revealed a non-conventional mechanism, the authors also established a classical function of miR-328 as a negative post-transcriptional regulator of *PIM1*, a molecule essential for the survival of CML [77]. miR-328 is a testament to the complexity of miRNA function as it is implicated not only in traditional regulatory processes but was also described as a decoy for an hnRNP. Restoration of base line miR-328 levels offers therapeutic promise as it addresses not only the differentiation block, but also may induce apoptosis of CML.

### 2.3. miRNAs Are Involved with TKI Resistance in BCR::ABL1 Independent Pathways

Yet, another type of clinically relevant miRNAs to CML is that which is directly linked to TKI resistance since those miRNAs dictate treatment strategies, predict outcomes, and offer promise for novel therapeutics. Each miRNA discussed here targets a different mRNA in different pathways, giving each a unique mechanism for therapy resistance.

Data shows that AKR1C3 is highly up-regulated in the resistant CML BMM, and when ectopically expressed, it decreased the efficacy of IM treatment in vitro [81]. Increases in AKR1C3 were sufficient to stimulate increased MAPK/ERK signaling. Conversely, miR-379-5p is downregulated in the BMM of CML but is capable of binding to AKR1C3 mRNA to suppress its translation. However, a recovery of miR-379-5p was sufficient to rescue the efficacy of IM by counteracting the increased AKR1C3 [81]. Similarly, miR-221 is decreased in mononuclear cells of peripheral blood taken from IM resistant compared to treatment sensitive CML patients [82]. The same study found that ectopic expression of miR-221 drove down the expression of STAT5. Previous studies validated STAT5 as a target of miR-221 [83], so the researchers sought to confirm the effect of aberrantly high levels of STAT5 due to a decrease of miR-221. They determined that STAT5 inhibited apoptosis while stimulating proliferation. Previous studies also demonstrated that STAT5 may be responsible for producing reactive oxygen species (ROS) in CML which contribute to therapy resistance by increasing genomic instability [84]. It is clear that miR-221 acts as a tumor suppressor in CML by reducing STAT5 expression; however, further studies are needed to elucidate all of its targets and effects. miR-153-3p is also downregulated in IM resistant CML patients and restoration of miR-153-3p re-sensitizes IM resistant CML while decreasing IM induced protective autophagy. A dual-luciferase assay confirmed Bcl-2 as the direct target of miR-153-3p. The decrease of miR-153-3p in resistant patients, and subsequent increase in Bcl-2, may attenuate therapy resistance, as CML studies have shown that increased Bcl-2 can decrease apoptosis and mediate therapy resistance [85]. Therefore, restoration of miR-153-3p in IM resistant patients may be an effective way to reestablish therapy response. Just as a decrease in miR-153-3p promotes protective autophagy of CML, so does the downregulation of miR-199a/b-5p that is seen in IM resistant K562 strains [86]. However, instead of targeting Bcl-2, miR-199a/b-5p targets WNT2. Ectopic expression of miR-199a/b-5p reduced WNT2 expression, prevented protective autophagy, and restored the apoptotic response of CML to IM therapy [86].

Alternatively, miR-577, which is downregulated in mononucleated cells in peripheral blood of CML patients, targets NUP160, which encodes part of a nuclear pore complex [87]. Previous studies established that nuclear core complexes may play a role in cancer survival by contributing to drug resistance and dormancy [88]. Dysregulated miR-577 leads to an increase in NUP160 which was shown to desensitize CML to IM, whereas, ectopic expression of miR-577 was able to rescue the efficacy of IM and restore proper regulation of NUP160 [87].

Researchers also determined miR-342-5p to be suppressed in CML patients while CCND1 expression levels were particularly high [89]. A luciferase assay confirmed that the 3′ UTR of CCND1 was a target of miR-342-5p [89]. Aberrant CCND1 expression has already been established as a contributor to the development of CML by promoting cell cycling [90]. According to the authors, miR-342-5p also acts as a tumor by suppressing *BCR::ABL1* transcripts indirectly [89]. Treatment with a miRNA mimic of miR-342-5p may potentially alleviate the increase of *BCR::ABL1* and CCND1 combatting therapy resistance.

Yet, another mechanism for drug resistance that plays an important role in CML is regulated by miRNAs: drug transporters. Efflux transporters allow cells to evade therapy by pumping the molecule out of the cytosol. The mononuclear cells of peripheral blood of IM resistant CML patients compared to responsive patients showed a decrease in miR-214 [91]. Interestingly, miR-214 binds to the 3′ UTR of ABCB1, which encodes for a drug efflux transporter known to be up-regulated in multi-drug resistant malignancies, although it does so with limited sequence homology [85,92]. Ectopic expression of miR-214 restored therapy response in IM resistant cell lines, making miR-214 a promising tool to address drug-resistance in CML [91].

An abundance of data indicates that miRNAs are dysregulated in CML, with a variety of mechanisms responsible for such dysregulation, including remodeled leukemic niches, epigenetic silencing, and activation or suppression of a variety of pathways. Once dysregulated, these miRNAs exhibit their effect on CML using diverse functions, including regulating *BCR::ABL1* transcripts, modulating the CML LSC, activating or suppressing pathways implicit in CML, and regulating drug transporters. The implementation of strategies to restore homeostatic expression of miRNAs, such as treatment with mimics or antagomiRs is needed in combination with highly specific tyrosine kinase inhibition to fully eliminate disease progression and relapse. Below in Table 1, we summarize miRNAs dysregulated in CML.

## 3. The Role of circRNAs in CML

Like miRNAs, circRNAs are ncRNAs which regulate a vast array of pathways and are frequently dysregulated in cancer and many other diseases. These single stranded RNA molecules are covalently linked at their 3′ and 5′ ends, generating a circular structure [93]. circRNAs are highly stable due to their structure, making them easily detectable in blood [94]. Because of their easy detectability and frequency of tissue specific dysregulation in disease, circRNAs offer promise as diagnostic and prognostic biomarkers [95,96,97]. A study of the circRNA Cdr1 as showed it contained over 70 binding site for the miR-7 [98]. Because of its stable, covalently bonded circular structure, circRNAs are not degraded by the miRNA which binds it [99]. Instead, the circRNA sequesters miRNAs, preventing them from acting on their downstream targets [98]. In this way, circRNAs control all the same processes in which miRNAs regulate. Here, we discuss the diagnostic and therapeutic potentials of several circRNAs which have been studied in the context of CML.

### 3.1. circRNA Are Generated by Translocation

Given that CML is the result of the chromosomal translocation which gives rise to the *BCR::ABL1* fusion gene, there is a clear potential for novel circRNAs to be generated by the fusion, as circRNAs are often found in coding regions of genes [97,100]. So far, two circRNAs resulting from translocations in CML have been identified and characterized. The first is circBA9.3, whose expression is linked to IM resistance. Data demonstrated that circBA9.3 promoted growth and IM resistance of CML; however, the mRNA levels of *BCR::ABL1* remained steady [101]. The authors concluded that circBa9.3 may be exerting its oncogenic effect by increasing translational efficiency of *BCR::ABL1* [101]. Previous studies have already established that circRNAs may regulate the genes in which they reside, such as CircPABPN1 and circFOXO3 [102,103]. However, F-circBA1, another circRNA formed by the fusion of *BCR::ABL1*, exerts an oncogenic effect by sponging miR-148-3p, which is responsible for limiting the expression of its target CDC25B thus regulating cell cycle progression [104]. When miR-148-3p is sequestered in F-circBA1, CDC25B becomes up-regulated, as seen in many cancers [105,106,107]. The presence of F-circBA1 was detected in 9/14 patient samples tested, indicating that many, but not all BCR/ABL1 positive patients express it [104]. Silencing of F-circBA1 initiated cell cycle arrest, indicating that F-cricBA1 could be a useful therapeutic target in patients who express it [104]. Since these circRNAs are generated from the fusion, which is responsible for the generation of CML, their expression has no “normal” biological function. Therefore, they should be targeted as completely as possible, and their suppression is unlikely to have any negative side effects. Furthermore, since their expression is not associated with any healthy biological function, they are excellent candidates for biomarkers in CML; their levels may correlate with presence of *BCR::ABL1* mRNA transcripts. Importantly, data indicates that F-circBA1 was more easily detectable than BCR::ABL1 transcripts following RNase R treatment, suggesting that F-circBA1 may serve as a more robust diagnostic marker than BCR::ABL1 transcripts [104].

### 3.2. circRNAs Are Increased in TKI Resistant Patients

Most circRNAs involved in CML therapy resistance are not generated from the classical translocation, but rather are dysregulated in CML. Circ_0009910 was shown to be consistently upregulated in IM resistant patients and was correlated with poor outcomes [108]. It was established that circ_0009910 knockdown could mitigate therapy resistance, and further study showed that miR-34a-5p, which regulates ULK1, is downregulated in resistant patients and is a target of the circRNA [108]. Therefore, ULK1 is highly expressed in IM resistant patients and induces autophagy, contributing to therapy resistance and making circ_0009910 a potential therapeutic target in IM resistant CML [108].

Similarly, hsa_circ_0058493 is deregulated in the mononucleated cells of peripheral blood of IM resistant CML patients and is associated with a worse prognosis [109]. Downregulation of hsa_circ_0058493 restored IM sensitivity in resistant cell lines, and bioinformatic analysis indicated the presence of two binding sites for miR-548b-3p [109]. Downregulation of has_circ_0058493 in vitro increased miR-548b-3p, indicating that it is likely sponged by hsa_circ_0058493 [109]. Together, the results indicate that hsa_circ_0058493 is a promising diagnostic and prognostic tool and potentially a future therapeutic target in IM resistant patients.

Unsurprisingly, some circRNAs contribute to therapy resistance in CML by exhibiting an effect on *BCR::ABL1* transcripts. One study showed that circ_0051886 and circ_0080145 are upregulated in K562 IM resistant cells and that they target miRNAs responsible for regulating the transcription of *ABL1* [110]. Knockdown of both circRNAs suppressed *BCR::ABL1* protein expression while overexpression of the circRNAs upregulated *BCR::ABL1* [110]. Thus, they established the circ_0051886/miR-637/ABL1 and circ_0080145/miR-203/ABL1 axes as important determinates of IM resistance in CML [110]. Another study demonstrated that circ_0080145 regulates the expression of *BCR::ABL1*; however, luciferase assay reported miR-29b as the target instead of miR-203 [111]. As mentioned in our discussion of miRNAs, miR-29b is capable of targeting *BCR::ABL1* mRNA transcripts and decreases proliferation while promoting apoptosis [49]. Yet, another target of circ_0080145 was identified as contributing to IM resistance in CML and the circ_0080145/miR-326/PPFIA1 axis was established as a regulator of IM resistance [112].

Clearly, circRNAs are dysregulated in CML but their roles in contributing to therapy resistance need to be studied further. The functions of circRNAs are complex, as they may act as competitive endogenous RNAs (ceRNA) against multiple miRNAs, each of which could potentially target multiple mRNA transcripts. Because of this, it is difficult to determine the exact mechanism of therapy resistance stemming from dysregulated circRNAs. Although many of these circRNAs may appear to possess therapeutic potential, due to their robust circular structure, they are highly stable and difficult to degrade. Instead, current application of circRNAs should be for biomarkers to predict outcomes and dictate treatment plans. Below in Table 2, we summarize circRNAs dysregulated in CML.

## 4. The Role of lncRNAs in CML

Any single stranded RNA which does not encode proteins and exceeds a length of 200 nucleotides is considered a lncRNA [113]. Like miRNAs and circRNAs, lncRNAs are responsible mainly for regulating gene expression which allows them to control a huge range of biological processes. Aberrant expression of lncRNAs is associated with many different cancers [114]. The diversity of the functions of lncRNAs is expanded compared to miRNAs and circRNAs because of the variety of roles lncRNAs play in the cell. Like miRNAs, complementary sequences present on lncRNAs empower them to interact with mRNA and recruit proteins which degrade it. Not only can lncRNAs lead to degradation of mRNAs, but they can also enhance the translation of mRNAs by stabilizing mRNA and promoting its association to ribosomes [115]. Like circRNAs, lncRNAs can also act as ceRNAs, sponging their target miRNAs and promoting the translation of those mRNAs targeted by those miRNAs. Further known functions of lncRNAs include modifying chromatin structure, mRNA splicing and transcription, signaling pathways, and nuclear organization [113]. The diverse skill set of lncRNAs can be attributed to their ability to interreact with other RNAs, DNA, and proteins regulating processes crucial for gene expression. Here, we explore lncRNAs known to be dysregulated in CML.

### lncRNAs Contribute to TKI Resistance in CML

TKI resistance presents a clinical challenge in the treatment of CML patients, necessitating further understanding of the processes which drive it. Due to the wide range of functions, they perform, lncRNAs can be upregulated or downregulated in CML. Maternally expressed 3 (MEG3) is a lncRNA which is downregulated in advanced stages of CML and was shown to be heavily methylated in IM resistant cells [116,117]. While two studies indicated miR-21 as a target of MEG3, one showed that overexpression of MEG3 in vitro decreases miR-21 promoting apoptosis and suggested that treatment with demethylating agents may be a potential therapeutic [116]. The other study confirmed that overexpression of MEG3 and or knock-down of miR-21 induces apoptosis, but also showed that it restored IM response by decreasing the expression of several drug transporters known to contribute to multi-drug resistance: MRP1, MDR1, and ABCG2 [117]. Yet, another study of MEG3 in CML showed increased methylation in CML patients when compared to healthy controls and suggested the use of the histone deacetylase inhibitor Chidamide to mitigate epigenetic silencing of MEG3 and miR-147 [118]. Interestingly, this study showed that HDACs were downregulated when MEG3 was overexpressed, indicating that the downregulation of MEG3 could contribute to hypermethylation of miR-147 [118]. In breast cancer, miR-147 suppressed phosphorylation of the PI3K/AKT pathway [119,120]. As mentioned in our discussion of miRNAs, inhibition of the PI3K/AKT pathway may rescue IM sensitivity in CML [54]. Thus, treatment with HDAC inhibitors may be an effective therapeutic strategy to restore MEG3 and miR-147 levels and limit unregulated PI3K/AKT signaling.

As one of the functions of MEG3 is to suppress a miRNA involved with the efflux of drugs, many other lncRNAs share the same function. For instance, the lncRNA SNHG5 and the drug efflux transporter ABCC2 were both dysregulated in CML, and their expression was correlated [121]. Functional assays confirmed that SNHG5 is in part responsible for drug resistance in K562 cell lines by sponging miR-205-5p, which negatively regulates ABCC2 [121]. Similarly, UCA1 is shown to act as a ceRNA against miR-16, increasing MDR1 expression, drug efflux, and ultimately therapy resistance [122]. The dysregulation lncRNA HOX antisense intergenic RNA (HOTAIR) also has been associated with an increase in drug transporters [123]. It is reported that HOTAIR is increased in multi-drug resistant CML patients and in IM resistant K562 cell lines [123]. Knockdown of HOTAIR in vitro conferred with IM sensitivity, decreased MRP1, and decreased PI3K/AKT phosphorylation [123]. Another study showed that HOTAIR acts as a ceRNA against miR-143 to promotes phosphorylation in the PI3K/AKT pathway, and that knockdown of HOTAIR or overexpression of miR-143 induced apoptosis [124]. They also demonstrated that aberrant HOTAIR expression is linked to DNMT1 and DNMT3A which are important for the epigenetic progression of CML, and that HOTAIR expression progressed in the transition from CP to BP [124,125]. It was reported that regulating HOTAIR limits its effect on binding DNMT1, thus preventing dysregulation of the tumor suppressor PTEN [126]. Together, these studies indicate that HOTAIR acts strongly as an oncogene and is a promising biomarker for IM resistance and therapeutic target.

Just as increased HOTAIR expression is associated with disease severity of CML, so is the lncRNA HULC. Knockdown of HULC led to an increase in the apoptosis inducing caspase-3 while lowering the expression of c-Myc and Bcl-2 [127]. It was also demonstrated that diminishing HULC led to a reduction in PI3K/AKT phosphorylation, restoring IM sensitivity in K562 cells. Previous studies reported targeting the PI3K/AKT pathway sensitizes CML LSCs to therapy, providing further support for the claim that HULC contributes to IM resistance by increasing phosphorylation of the PI3K/AKT pathway [128].

Some lncRNAs are clearly dysregulated in CML, but either no or conflicting reports on their functions in CML exist. For instance, lncRNA nuclear-enriched abundant transcript 1 (NEAT1), which plays a role in nuclear structure and scaffolding, is downregulated in CML [129]. NEAT1 is negatively regulated by the oncogene c-MYC, which is downstream of *BCR::ABL1* and binds to its promoter, explaining why NEAT1 is suppressed in CML [129]. Thus, *BCR::ABL1* inhibition restores NEAT1 expression, conferring with IM resistance. NEAT1 is associated with IM sensitivity due to its ability to bind paraspeckle protein splicing factor proline/glutamine-rich (SFPQ), which is involved in apoptotic pathways [129]. However, another study demonstrated that overexpression of NEAT1 contributed to apoptosis and inhibition of proliferation by sponging miR-766-5p, which targets the cell-cycle regulator cyclin-dependent kinase inhibitor 1A (CDKN1A) [130]. Although, NEAT1 is downregulated in CML, the function of NEAT1 in CML has not yet been well established and further studies could reveal the diagnostic and therapeutic potential of NEAT1. While there are contradictory studies on the function of NEAT1 in CML, no studies have yet explored the role of CCAT2 in CML, although one has shown lncRNA CCAT2 is dysregulated in CML, with unusually high expression in the CP of CML compared to healthy controls. Thus, it offers promise as biomarker for CML, but further studies could reveal a therapeutic potential as well [131].

Due to the complex function of lncRNAs, it is difficult to fully elucidate the mechanisms in which they regulate CML therapy resistance. Further efforts are needed to establish lncRNAs as biomarkers and therapeutic targets, but here we have established that the dysregulation of lncRNAs in CML resistance warrants further study. Below in Table 3, we summarize lncRNAs dysregulated in CML.

## 5. Conclusions

The development of TKIs drastically changed the treatment and prognosis of CML for countless patients. However, IM resistance and TKI resistance in general, remains an obstacle to curing CML. Because of this, the role of ncRNAs in therapy resistance has been explored and offers promise both as biomarkers and therapeutic targets. Remodeling of the BMM, gene fusions, epigenetic modifications, pathway amplifications, and many other mechanisms are responsible for the dysregulation of ncRNAs in CML. Aberrantly expressed ncRNAs then activate or suppress molecular pathways implicit in therapy resistance, upregulate drug transporters, regulate *BCR::ABL1* expression, and modulate the CML LSC to contribute to therapy resistance in CML. The various classes of ncRNAs offer promise as potential biomarkers or therapeutic targets. Further research is clearly necessary before the potential of these ncRNAs can be fully realized, but this review has established that ncRNAs play an important role in the development of CML therapy resistance and warrant further study. Below in Figure 1, we summarize mechanisms in which dysregulated ncRNAs contribute to CML.

## Figures and Tables

**Figure 1 ijms-23-12271-f001:**
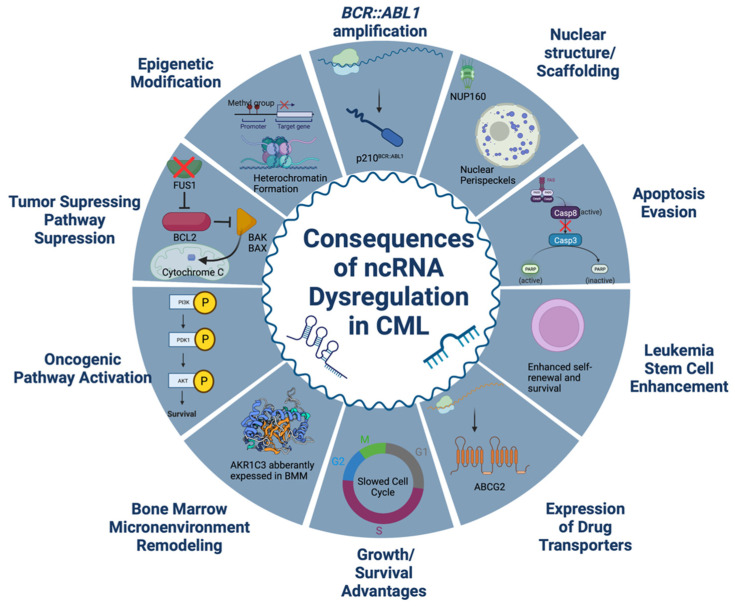
Non-coding RNAs dyregulated in CML exert their effect through a variety of mechanisms. Created with BioRender.com.

**Table 1 ijms-23-12271-t001:** Summary of discussed miRNAs dysregulated in CML.

miRNA	Dysregulation in CML	Function	Dysregulation Consequence	Relevance	Citation
miR-29b	Downregulated	Targets 3′ UTR of *BCR::ABL1*	Increased growth/survival	Therapeutic	[49]
miR-30a	Downregulated	Suppresses Beclin 1 and ATG5	Decreased apoptosis, increased *BCR::ABL1*	Therapeutic	[50,52]
miR-424	Downregulated	Targets 3′ UTR of *BCR::ABL1*	Reduced apoptosis, enhanced growth, desensitized to IM	Therapeutic	[51]
miR-320a	Downregulated	Targets *BCR::ABL1*	Increased phosphorylation of the *PI3K*/*AKT* pathway	Therapeutic	[53]
miR-196b	Downregulated	Targets *BCR::ABL1* and HOXA9	Increased cell cycling and growth	Therapeutic	[55]
miR-23a	Downregulated	Regulates *BCR::ABL1*	Increased *BCR::ABL1* and disease progression	Therapeutic	[56]
miR-96	Downregulated	Binds 3′ UTR of *BCR::ABL1*	Transition from CP to BP	Diagnostic	[60]
Let-7	Downregulated	Increases LIN28B	Enhanced LSC self-renewal and transformation	Therapeutic	[63]
miR-203	Downregulated	Targets 3′ UTRs of survivin and Bmi-1 transcripts	Enhanced LSC growth and renewal	Therapeutic	[64]
miR-494-3p	Downregulated	Targets 3′ UTR of c-MYC	Increases survival of LSCs	Therapeutic	[67]
miR-21	Upregulated	Activates PI3K/AKT Pathway	Enhances LSC survival in the presence of IM	Therapeutic	[54,69]
miR-29a-3p	Upregulated	Targets 3′ UTR *TET2*	Decreased apoptosis of LSCs	Therapeutic	[70]
miR-660-5p	Upregulated	3′ UTR *EPAS1*	Protects LSCs	Therapeutic	[67]
miR-378	Upregulated	Increases Nanog, Oct4, and c-MYC	Promotes stem cell pluripotency	Therapeutic	[71]
miR-126-3p	Upregulated in endothelial cells	Regulates LSC dormancy and engraftment potential	Increases therapy resistance	Therapeutic	[74]
miR-328	Downregulated in CML BP	Acts as decoy against hnRNP E2, allowing *CEBPA* ExpressionTargets 3′ UTR of *PIM1*	Differentation blockIncreased survival	Therapeutic	[77]
miR-379-5p	Downregulated	Targets *AKR1C3* mRNA	Increases therapy resistance	Therapeutic	[81]
miR-221	Downregulated in IM resistant patients	Targets STAT5 mRNA	Increases survival and decreases apoptosis	Therapeutic	[83]
miR-153-3p	Downregulated in IM resistant patients	Targets Bcl-2	Reduces IM induced apoptosis	Therapeutic	[85]
miR-199a/b-5p	Downregulated in IM resistant K562	Targets WNT2	Promotes protective autophagy	Therapeutic	[86]
miR-577	Downregulated	Targets NUP160	Desensitizes CML to IM	Therapeutic	[87,88]
miR-342-5p	Downregulated	Targets CCND1	Promotes cell cycling, increases *BCR::ABL1*, confers therapy resistance	Therapeutic	[89]
miR-214	Downregulated in IM Resistant Patients	Targets 3′ UTR of ABCB1	Decreases drug efflux transporter	Therapeutic	[91]

**Table 2 ijms-23-12271-t002:** Summary of discussed circRNAs dysregulated in CML.

circRNA	Dysregulation in CML	Function	Dysregulation Consequence	Relevance	Citation
circBA9.3	Generated by translocation	Complicit in translation of *BCR::ABL1*	Therapy Resistance	Therapeutic/Diagnostics	[101]
F-circBA1	Generated by translocation	Sponges miR-148-3p which targets *CDC25B*	Increased cell cycling	Therapeutic	[104]
Circ_0009910	Upregulated in IM resistant patients	Targets miR-34a-5p which regulates ULK1	Increases protective autophagy	Therapeutic	[108]
hsa_circ_0058493	Upregulated in IM resistant patients	Sponges miR-548b-3p	Therapy Resistance	Therapeutic/Diagnostic	[109]
circ_0051886	Upregulated in IM resistance K562	Targets miR-637	Increases *BCR::ABL1* Translation	Therapeutic/Diagnostic	[110]
circ_0080145	Upregulated in IM resistance K562	Targets miR-203	Increases *BCR::ABL1* Translation	Therapeutic/Diagnostic	[110]

**Table 3 ijms-23-12271-t003:** Summary of discussed lncRNAs dysregulated in CML.

lncRNA	Dysregulation in CML	Function	Dysregulation Consequence	Relevance	Citation
MEG3	Downregulated in advanced stages of CML	Targets miR-21	Increased drug transporters, Therapy Resistance	Therapeutic	[101,102]
SNHG5	Downregulated in IM resistant K562	Sponges miR-205-5p	Increased ABCC2, therapy resistance	Therapeutic	[106]
UCA1	Increased in IM Resistant Cell Line	ceRNA against miR-16	increases MDR1, drug efflux, and therapy resistance	Therapeutic/diagnostic	[107]
HOTAIR	Increased in multidrug resistant patients	ceRNA against miR-143	promotes phosphorylation of the *PI3K*/*AKT* pathway	Therapeutic and diagnostic	[109,110]
HULC	Increased expression correlates with disease stage	increasing phosphorylation of the *PI3K*/*AKT* pathway	Therapy Resistance	Diagnostic	[127,128]
NEAT1	Downregulated	Unknown	Unknown	Diagnostic	[129,130]
CCAT2	Upregulated in BP	Unknown	Unknown	Diagnostic	[131]

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
