# Peer review of "Non-Coding RNAs Are Implicit in Chronic Myeloid Leukemia Therapy Resistance"

_ijms, 2022, doi:10.3390/ijms232012271_

Round 1

Reviewer 1 Report

This is a very nice review article that provides the reader with a comprehensive view of the possibility of post-transcriptional regulation by non-coding forms of RNA interfering with the onset and development of CML. The article is written logically, comprehensibly and provides a range of information necessary for orientation in the issue. In principle, I have no strong objections to this article, I just give the authors something to consider:
1. Is it appropriate to call CML a rare disease when, according to the American Cancer Society (https://www.cancer.org/cancer/chronic-myeloid-leukemia/about/statistics.html), about 15% of all new cases of leukemia are chronic myeloid leukemia and about 1 in 526 people will get CML in their lifetime in the United States?
2. It would be appropriate to state in the article that BCR-ABL is not limited to CML. It also occurs in 11%–29% of patients with ALL, but is relatively rare in childhood ALL (1%–3%) (reviewed in doi.org/10.1186/s40880-016-0108-0).
I must declare here that I have no affiliation with this article or any of its authors.

Author Response

  1. Is it appropriate to call CML a rare disease when, according to the American Cancer Society (https://www.cancer.org/cancer/chronic-myeloid-leukemia/about/statistics.html), about 15% of all new cases of leukemia are chronic myeloid leukemia and about 1 in 526 people will get CML in their lifetime in the United States?

Author response: Thank you for pointing this out. The reviewer is correct, and we have removed the statement that CML is a rare disease from our draft.

The revised text reads as follows on [line 25]:

“[Chronic myeloid leukemia (CML) is a myeloproliferative disorder cytogenetically recognizable by the presence of the Philadelphia (Ph) chromosome.]”

  1. It would be appropriate to state in the article that BCR-ABL is not limited to CML. It also occurs in 11%–29% of patients with ALL, but is relatively rare in childhood ALL (1%–3%) (reviewed in doi.org/10.1186/s40880-016-0108-0).

Author response: Thank you for pointing this out. We agree that this is appropriate information to include in our review and have added it.

The revised text reads as follows on [line 36]:

“[In addition to CML, p210BCR-ABL1 transcripts have also been rarely detected in Acute Lymphocytic Leukemia (ALL) and Acute Myeloid Leukemia (AML)[15, 16]. p210BCR-ABL1 is the most common product of the fusion in CML but a shorter transcript p190BCR-ABL is also detected in CML, although more commonly in AML and B-cell ALL (B-ALL)[15, 16]. A third, larger variant p230BCR-ABL1  is the major diagnostic marker for Neutrophilic-chronic myeloid leukemia (CML-N)[17].]”

Reviewer 2 Report

This is a very well written overview of non-coding RNAs reported in CML.

1.      Line 165. Introducing hypomethylating agents could potent potentially restore the function of some miRNAs. Could the authors discuss whether such agents have been used in CML and their efficacy.

2.      Line 199. miRNA mimics is a ‘promising therapeutic strategy’. At this stage in the development of such agents a more appropriate term could be ‘potential therapeutic strategy’.

3.      Line241. …differentiation is an established obstacle in both AML and CML. Should this read blast phase CML, rather than CML in general.

4.      Please expand on the F-circBA1, which is listed as another circRNA formed by the fusion of BCR::ABL1. How frequently has this circRNA been reported in patient samples? Is it exclusively detected in patients that express BCR::ABL1? Is it associated with TKI resistance?

5.      Line 362 suggests that circRNAs are excellent candidates for biomarkers for CML and may be easier to detect than BCR::ABL1. Is there any evidence that circRNAs are easier to detect than BCR::ABL1?

Minor points

1.      The current recommended gene fusion nomenclature is BCR::ABL1 and should be used throughout.

2.      BCR::ABL1 not only generates a p210 protein as indicated in the first paragraph. p210 is the most common in CML. TKIs not only inhibit p210 but also other forms of BCR::ABL1.

3.      A few typographical errors: therefor instead of therefore. Reference 87 in the reference list.

Author Response

Reviewer 2

  1. Line 165. Introducing hypomethylating agents could potent potentially restore the function of some miRNAs. Could the authors discuss whether such agents have been used in CML and their efficacy.

Author response: Thank you for introducing this question. We agree that this is appropriate information to include in our review and have added it.

The revised text reads as follows on [line 196]:

“[Indeed, the use of hypomethylating agents to treat CML has been documented, with recent data indicating a synergistic effect of an oral demethylating agent (OR-2100) with TKIs on CML growth in-vitro[57]. Another study demonstrated some success in treating IM resistant patients with low doses of decitabine[58]. It was also demonstrated that the combination of administration of low doses of decitabine along with IM is effective in some patients with advanced stages of CML[59].]”

  1. Line 199. miRNA mimics is a ‘promising therapeutic strategy’. At this stage in the development of such agents a more appropriate term could be ‘potential therapeutic strategy’.

Author response: We agree that it would be most appropriate to describe miRNA mimics as a potential therapeutic strategy at this point in time.

The revised text reads as follows on [line 245]:

“[Since these miRNAs help suppress stemness in CML or sensitize the stem cell to TKI treatment, restoring their expression in patients using miRNA mimics is a potential therapeutic strategy in combination with traditional therapy.]”

  1. Line241. …differentiation is an established obstacle in both AML and CML. Should this read blast phase CML, rather than CML in general.

Author response: Yes, thank you.

The revised text reads as follows on [line 287]:

“[Aside from the treatment challenges presented by LSCs, impaired differentiation is an established obstacle in both AML and BP CML[76, 77].]”

  1. Please expand on the F-circBA1, which is listed as another circRNA formed by the fusion of BCR::ABL1. How frequently has this circRNA been reported in patient samples? Is it exclusively detected in patients that express BCR::ABL1? Is it associated with TKI resistance?

Author response: Thank you. These are important questions to address in the review. The source indicated that F-circBA1 is expressed in patient samples (9/14 tested). The authors imply, but do not directly state that this circRNA is expressed only in BCR::ABL1 positive CML patients. They do indicate that some BCR::ABL1 patients do not express F-circBA1. The authors do not indicate whether F-circBA1 is associated with therapy resistance, however, it would be important for future studies to explore this.

The revised text reads as follows on [line 424]:

“[The presence of F-circBA1 was detected in 9/14 patient samples tested, indicating that many, but not all BCR/ABL1 positive patients express it[105].]”

  1. Line 362 suggests that circRNAs are excellent candidates for biomarkers for CML and may be easier to detect than BCR::ABL1. Is there any evidence that circRNAs are easier to detect than BCR::ABL1?

Author response: Yes, the authors of source #105 showed that following RNAse R treatment, BCR/ABL1 transcripts were no longer detectable but F-circBA1 was. We agree that this is important evidence to include!

Tan, Y.; Huang, Z.; Wang, X.; Dai, H.; Jiang, G.; Feng, W., A novel fusion circular RNA F-circBA1 derived from the BCR-ABL fusion gene displayed an oncogenic role in chronic myeloid leukemia cells. Bioengineered 2021, 12, (1), 4816-4827.

The revised text reads as follows on [line 433]:

“[ Importantly, data indicates that F-circBA1 was more easily detectable than BCR::ABL1 transcripts following RNase R treatment, suggesting that F-circBA1 may serve as a more robust diagnostic marker than BCR::ABL1 transcripts[105].

Minor points

  1. The current recommended gene fusion nomenclature is BCR::ABL1 and should be used throughout.

Author response: Thank you for pointing this out. According to HUGO Gene Nomenclature Committee (HGNC) the current recommended gene fusion nomenclature is BCR::ABL1. We have corrected this within the draft.

  1. BCR::ABL1 not only generates a p210 protein as indicated in the first paragraph. p210 is the most common in CML. TKIs not only inhibit p210 but also other forms of BCR::ABL1.

Author response: For the sake of clarity, this was omitted from the original draft. However, we do agree that this would be an important thing to include.

The revised text reads as follows on [line 36]:

“[In addition to CML, p210BCR-ABL1 transcripts have also been rarely detected in Acute Lymphocytic Leukemia (ALL) and Acute Myeloid Leukemia (AML)[15, 16]. p210BCR-ABL1 is the most common product of the fusion in CML but a shorter transcript p190BCR-ABL is also detected in CML, although more commonly in AML and B-cell ALL (B-ALL)[15, 16]. A third variant p230BCR-ABL1  is the major diagnostic marker for Neutrophilic-chronic myeloid leukemia (CML-N)[17].]”

The revised text reads as follows on [line 67]:

“[The current first-line standard of care for CML patients is treatment with tyrosine kinase inhibitors (TKIs) that competitively or allosterically inhibit p210BCR-ABL1 ,as well as other BCR::ABL1 variants, although less effectively[2, 20-22]. ]”

  1. A few typographical errors: therefor instead of therefore. Reference 87 in the reference list.

Author response: Thank you, the draft has been corrected accordingly.